# Long-Term Population Dynamics of Namib Desert Tenebrionid Beetles Reveal Complex Relationships to Pulse-Reserve Conditions

**DOI:** 10.3390/insects12090804

**Published:** 2021-09-08

**Authors:** Joh R. Henschel

**Affiliations:** 1South African Environmental Observation Network, P.O. Box 110040 Hadison Park, Kimberley 8301, South Africa; joh.henschel@gmail.com; 2Centre for Environmental Management, University of the Free State, P.O. Box 339, Bloemfontein 9300, South Africa; 3Gobabeb Namib Research Institute, P.O. Box 953, Walvis Bay 13103, Namibia

**Keywords:** tenebrionidae, darkling beetle, long term ecological research, population signature, irruption, population crash, rainfall, nonrainfall moisture, detritivore, species diversity

## Abstract

**Simple Summary:**

Rain seldom falls in the extremely arid Namib Desert in Namibia, but when a certain amount falls, it causes seeds to germinate, grass to grow and seed, dry, and turn to litter that gradually decomposes over the years. It is thought that such periodic flushes and gradual decay are fundamental to the functioning of the animal populations of deserts. This notion was tested with litter-consuming darkling beetles, of which many species occur in the Namib. Beetles were trapped in buckets buried at ground level, identified, counted, and released. The numbers of most species changed with the quantity of litter, but some mainly fed on green grass and disappeared when this dried, while other species depended on the availability of moisture during winter. Several species required unusually heavy rainfalls to gradually increase their populations, while others the opposite, declining when wet, thriving when dry. All 26 beetle species experienced periods when their numbers were extremely low, but all had the capacity for a few remaining individuals to repopulate the area in good times. The remarkably different relationships of these beetles to common resources, litter, and moisture, explain how so many species can exist side by side in such a dry environment.

**Abstract:**

Noy-Meir’s paradigm concerning desert populations being predictably tied to unpredictable productivity pulses was tested by examining abundance trends of 26 species of flightless detritivorous tenebrionid beetles (Coleoptera, Tenebrionidae) in the hyper-arid Namib Desert (MAP = 25 mm). Over 45 years, tenebrionids were continuously pitfall trapped on a gravel plain. Species were categorised according to how their populations increased after 22 effective rainfall events (>11 mm in a week), and declined with decreasing detritus reserves (97.7–0.2 g m^−2^), while sustained by nonrainfall moisture. Six patterns of population variation were recognised: (a) increases triggered by effective summer rainfalls, tracking detritus over time (five species, 41% abundance); (b) irrupting upon summer rainfalls, crashing a year later (three, 18%); (c) increasing gradually after series of heavy (>40 mm) rainfall years, declining over the next decade (eight, 15%); (d) triggered by winter rainfall, population fluctuating moderately (two, 20%); (e) increasing during dry years, declining during wet (one, 0.4%); (f) erratic range expansions following heavy rain (seven, 5%). All species experienced population bottlenecks during a decade of scant reserves, followed by the community cycling back to its earlier composition after 30 years. By responding selectively to alternative configurations of resources, Namib tenebrionids showed temporal patterns and magnitudes of population fluctuation more diverse than predicted by Noy-Meir’s original model, underpinning high species diversity.

## 1. Introduction

Desert ecosystems appear to lack stability, rendering it difficult to define equilibrium points [1]. However, Noy-Meir [2,3] posited that, paradoxically, desert ecosystems exhibit remarkable stability by undergoing rapid and repeatable irruptions in response to erratic, rare rainfall events, then subsiding until the next effective rainfall. This insight gave rise to pulse-reserve models, which have subsequently been expounded [4]. However, protracted reserve phases, which are the norm in arid regions, have rarely been elucidated. Notwithstanding records on vertebrates, such as birds, mammals, frogs and reptiles [5,6], relatively few studies have been concerned with arthropod populations that bridge interpulse periods while maintaining the capacity to irrupt in response to sudden increases in water and food availability. Even extensively studied locusts have received far more attention during the pulse (gregarious swarming) phase than during the reserve (solitary resident) phase, even though the latter is known to affect the characteristics of the pulse [7].

Detritus, comprising dead plant and animal litter, is an important component of the reserve in deserts, concentrated and buried in certain locations and regularly remobilised by wind [8]. It stabilises energy flux and increases consumer populations, food web complexity, and species diversity [9]. The decomposition of detritus depends on its composition [10] and is affected by abiotic factors, such as UV photodegradation, temperature, and soil mixing [11], modified by habitat structure [12]. It is consumed by micro and macrodetritivores, whose activities are modified by above and below ground moisture and temperature conditions [13]. Detritivores increase in abundance and persist long after the rain’s moisture has disappeared, and the steady decomposition of detritus [14,15] leads to gradual population decline as food becomes limiting [16].

The Namib Desert, one of the driest and least productive deserts globally [17,18], erratically experiences productivity pulses. Effective rainfalls activate hypolithic and soil microbes that fix nitrogen [19], recycle nutrients from detritus through fungal decomposition [20,21], germinate seeds [22], and support the grass growth [23,24] that generates fresh detritus. This local production is supplemented by windblown allochthonous detritus [25,26,27].

The detritivore assemblage in the Namib is richer, in both species diversity and patterns of habitat adaptation, than other extreme deserts (MAP < 50 mm), such as the Sahara, Empty Quarter, and Atacama [12,25,28]. There are over 200 species of tenebrionid beetles (Coleoptera: Tenebrionidae) in the Namib Desert [25]. Although, in general, they are euryphagous opportunists, in effect, most of them are dependent upon various components of detritus [16,25,26,28,29]. The diversity of some other Namib detritivores, such as termites [30], zygentomans [31], and collembolans [32], is also remarkably high, as is that of their predators, such as solifuges, scorpions, and spiders [33]. This richness illustrates another paradox: trophic structures and diversity are more complex than expected based on productivity [34,35], well expressed in the Namib [36].

Near surface moisture represents another important feature of the reserve. Nonrainfall moisture (NRM), comprising fog, dew, atmospheric vapour, and groundwater evaporates, are briefly available during many nights in the Namib [37,38]. Tenebrionids can extract moisture from these sources, imperative for adult survival, egg production and hatching, and larval development [39]. Although NRM does not increase the quantity of detritus, its moisture and activation of microbes (including ecoholobionts) [40,41,42] may improve digestibility to tenebrionids. In addition, water from rainfall can linger in the upper soil for several weeks, depending on the amount of rainfall, the nature of the soil, and the season [43,44], providing longer opportunities for the processes mentioned above to increase the reproductive success and population densities of tenebrionids following a pulse [45,46,47,48,49].

During dry periods without pulses, reproductive output does not keep up with mortalities from predation [35,50], dehydration, starvation, and age, so populations gradually decline. Not having dormant eggs or larvae, the long lived adult tenebrionids (with a lifespan of several years [51,52]) represent the active reserve, maintained by accessible moisture and food [39]. Although the life history, activity patterns, and phenology of tenebrionids vary [53,54,55,56], many Namib species are ready to reproduce whenever they obtain sufficient resources [57,58], enabling them to respond quickly to pulses.

The current study elucidates Noy-Meir’s [2,3] premise using a long term data series of tenebrionid populations occurring on the Central Namib gravel plain. The population changes of 26 tenebrionid species were tracked over 45 years (1976–2020), the first half of this period being considerably drier than the second [18], representing a natural climate change experiment. This paper tests the hypothesis that the magnitudes and durations of population increases are related to the effects of irregular pulses of effective rainfall and that population persistence is related to the availability of moisture and detritus, the reserve. It is predicted that sympatric tenebrionid species adopt different population trajectories in tracking resource pulses and reserves, influenced by particulars of their life history, activity patterns, and phenology. Lastly, this paper deliberates on the rationale and operability of such labour intensive institutional long term ecological research.

## 2. Materials and Methods

Observations were made between 1976 and 2020 near Gobabeb (23°33′ S, 15°02′ E) in the Namib Desert, about 1000 km from its southern and northern extremes and 60 km from its western and eastern boundaries. The ephemeral Kuiseb River separates the 40,000 km^2^ Namib Sand Sea dune field from the 40,000 km^2^ Central Namib Desert gravel plain.

The data analysed here were collected at a gravel plain site located 2 km from Gobabeb [59]. Data from five other habitats monitored for several decades, namely, an interdune plain, dune slope, dune slipface, interdune hummocks, and a riverbed, entail different analytical approaches than used in the current paper and are not presented here. The gravel plain site is flat, firm, and usually bare of plants, except when overgrown with ephemeral grasses following effective rain events (Figure 1). Detritus is trapped under stones and in soil depressions, places favoured by tenebrionids. The study site, littered with quartz and granite pebbles, lies adjacent to a rocky ridge and a shallow drainage line within 500 m, with the riverbed of the ephemeral Kuiseb River 2 km distant.

From 1963–2020, rainfall data were collected at the Gobabeb First Order Weather Station. Tropical temperate troughs, mainly fed with moisture from Angola Low, are the main drivers of rain over this part of the Namib in late summer and early winter, January–May [18]. This system is distinct from the westerly cold fronts of the winter rainfall region of the Cape West coast and southern Namib.

Data on fog deposition were collected at the same site from 1966–2020, with seven years of missing data. Until 2003, fog precipitation was measured with a Grunow cylinder [60], until 2011 with a Schemenauer–Cereceda screen [61], and from 2014 onwards with a Juvik cylinder [62]. The three datasets were connected by assuming that the averages were comparable. Accordingly, data were transformed to Juvik Units (JU) by multiplying Grunow units by 3.3 and Schemenauer–Cereceda units by 0.01. Years were divided into two seasons: summer, October to March, and winter, April to September. In winter, soil temperatures at 10 cm depth were 3–9 °C cooler, and daily evapotranspiration was about 3 mm less than in summer [60].

Effective rainfall, defined as >10.9 mm within a week, stimulates grasses to germinate, flower, seed, and then add to the detritus pool [23,24]. Beyond the minimum threshold, additional rain that fell during the same season was considered part of the same event, as it watered growing plants. The detritus pool declines as it decomposes due to photodegradation and consumption by fungi, microbes, and invertebrates. Decomposition is affected by the frequency, quantity, and duration of near surface soil moisture following rainfall and NRM events [20,41].

Initial ballpark figures derived from the grass productivity and decomposition studies mentioned above are applied to derive a crude index of the detritus pool to reflect its general size, notwithstanding rainfall, productivity, and decomposition being heterogeneous at multiple spatial scales [63], partly allochthonous, and different between habitats. This index allows tracking the effect of rainfall over time through its effects on grass productivity and the decay of detritus. The quantity of detritus was calculated as the sum of annual productivity and decomposition rates, assuming a linear relationship: *Dj* = *Pj* + *a* × *Dj* − 1
where *D* is the index of the grass detritus pool, *j* is the current year, and *j −* 1 the previous year, *P* is the productivity of grass following effective rainfall [23,24]: *P* = 0.5872 × (*effective rainfall* − 10.93)
and *a* is the fraction of previous detritus remaining after annual decomposition, being 84% without rainfall [20] and decreasing with rainfall: *a* = 0.84 − (*annual rainfall*/100).

Pitfall traps 15 cm in diameter were deployed in March 1976 (the project is ongoing). At the gravel plain study site (23°32.628′ S, 15°02.973′ E), 15 traps were placed in five groups of three, 0.4–6.75 m apart within groups, and 50–200 m between [59,64]. Traps were monitored twice or thrice weekly for 90.5% of the time (221,355 trap-days). Solifugids, scorpions, spiders, lizards, and geckos, amounting to 4.3% of captures, could have preyed on trapped beetles, as could have transient birds and gerbils, but their impact on the results is unknown, and is assumed not to have biased the relative records of tenebrionid species over time. Live trapped tenebrionids were usually identified at the traps, alternatively in the laboratory, and then released a short distance from the traps. During the first 25 years, taxonomically verified voucher specimens guided the identification of species. Later, when this project was incorporated into the institutional training program, a photo album of dorsal or lateral images was used as an identikit. Previous records guided the final identification [26,59,64,65,66,67,68]. Data analyses were performed only on the most regularly trapped species, recorded >120 times [69], referred to as focal species (Appendix A, Table A1).

“Abundance”, here, refers to the capture rate, a function of trappability, density, and activity. The annual time scale encompasses weather related fluctuations of activity. Annual abundance data were standardised to captures per full trap-years to analyse trends, rounded to whole numbers, keeping singletons constant.

Following Wolda [70], arithmetic differences in annual abundance (*N*) between successive years were expressed by the gradation coefficient (*GC*):*GC* = log(*Nj*) − log(*Nj* − 1)
where *Nj* is *N* + ½ in year *j* and *Nj* − 1 for the year before year *j*. At low annual abundance (*N* < 20), caution is necessary when interpreting *GC* values, as trapping becomes increasingly sporadic. The annual variation coefficient, AV, is the variance of all *GC* values. *GC* and AV were determined for all study species and all successive years. Abundance fluctuations were tested for cyclicity by autocorrelation (lagged 3–22) of log(*N* + ½) and *GC*. Finally, species were compared by correlation (Pearson’s r), which formed the basis of the distance measures for a cluster analysis, using Ward’s method of hierarchical grouping attained by minimising the sum of squares [71]. Species that correlated and clustered most closely (*p* < 0.001) were assigned to distinct groups.

The years when the populations of each species began to increase were examined for rainfall events in that and the previous year to identify any connection between abundance and rainfall. Then, monthly trends [69] were examined for details of responses. A population was identified as having responded to rain (“triggered”) when *GC* increased by >0.05 (≈10% increase in N), provided that N increased by >10 (i.e., fluctuations at very low abundance, such as a doubling from 1 to 2, were discounted). Population irruptions are here defined as year to year increases in abundance by over 200% (*GC* > +1.3), and population crashes as year to year decreases over 200% (*GC* < −1.3). The initial response time is the period (months) that elapsed before a sustained population increase was detectable after a rainfall event. One-year lags were tested in the same way to account for larval maturation in seasonally active species. Rainfall over a week that triggered population increases was categorised as follows: heavy rainfall (>40 mm per annum), effective rainfall (>10.9 mm per week), light rainfall (<10.9 mm per month), summer, and winter rainfall. Decreases in abundance over an order of magnitude from maxima marked the end of responses. 

Changes in the annual composition of communities at each site were determined on square-root transformed abundance data of each focal species. From the yearly abundance table of species, a resemblance matrix was calculated based on Bray–Curtis similarities between all the pairwise combinations of years for each ecosystem. This matrix was processed with the nonparametric Mantel tests (Spearman R) [72] with 9999 permutations against two null model matrices: (1) the seriation matrix codes for equidistant steps between consecutive years (e.g., 1978 vs. 1979 = 1 year distance, 1978 vs. 1980 = 2 years distance…); (2) the cyclicity matrix codes for the resemblance of community structure at different times of the sampling period.

Data were analysed using Statistica 7.1 (2005, StatSoft Inc., Tulsa, OK, USA). Unless stated otherwise, *p*-values were less than 0.05. 

## 3. Results

### 3.1. Rainfall, Fog, and Detritus

Between 1963 and 2020 at Gobabeb, the mean annual rainfall was 25.3 ± SD 33.0 mm (0–171.9 mm; median 12.2 mm) (Table A2). Annual rainfall was highly variable (CV = 130%) and followed no discernible pattern across years (Figure 2a) (autocorrelation with lags of 3–22 y: r < 0.05; *p* > 0.1). In one year (1992), no rain fell; in three other years, annual rainfall was 1 mm or less (Table A2). Seasonal rainfall was not correlated (r = 0.12, *p* > 0.05), with most (59%) rain falling in late summer (January–March) and 27% in early winter (April–May) (Figure A1). Winter rainfall was weakly cyclical at 5 y intervals (r = 0.35).

Effective rainfall pulses were recorded in 23 of the 58 years, occurring at intervals of 2.3 ± SD 2.0 y (Table A2, Figure 2b) and were correlated with annual rainfall (r = 0.81), as well as summer (r = 0.69) and winter (r = 0.39) rain, with no autocorrelation. Three intervals between pulses before 1997 were 7–8 years long, while all other intervals were <3 y (Table A2). Seven heavy effective rainfalls (Q4, >40 mm) occurred in three groups, 1976–1978, 1997, and 2006–2011. The pitfall trapping of tenebrionids began in the year (1976) of the first heavy rainfall. During the first 21 years of trapping, there were seven effective rainfall events (MAP = 19.6 mm), compared to 15 in the 24 years that followed (MAP = 35.3 mm). The longest interval between heavy rainfalls during the first period was 19 years (6838 days between the last rain of the 1978 event and the first rain of the 1997 event).

Fog precipitation was measured in 47 of the years, annually recording 118 ± SD 56 Juvik units. Annual fog and rain precipitation were not correlated (r = 0.085), but fog was weakly autocorrelated at 21-year intervals (r = 0.24). Annual fog precipitation was significantly higher in the 18 years, 1979–1996, between heavy rainfall events than in the other 20 years, during periods with more rain (150 ± SD 52 vs. 96 ± SD 57 Juvik units, t = 3.044, *p* < 0.005; Figure 2c). 

The calculated detritus reserve reflected large productivity pulses following heavy rainfalls, with the highest increases in 1976, 1997, 2006, 2011, and 2018 (Figure 2d). The reserve gradually declined between 1978 and 1996, besides a minor increase in 1989–1990. After a boost in 1997, detritus declined until 2006. It reached an overall peak in 2011, with a subsequent small increase in 2018 (Figure 2d).

### 3.2. Trapping Data Characteristics

In total, 63,291 captures of tenebrionids were made at a rate of 0.31 ± SD 0.43 trap^−1^ day^−1^ (annual range 0.0016–2.4644 trap^−1^ day^−1^). The lowest rate was one tenebrionid captured in 15 continuously deployed traps during 309 days in 1994, while the maximum captured in a single trap in three days was 134 in 2011. Peak trap rates were recorded in 1976/77, 2000, 2006, 2008, 2011, and 2018, with each peak associated with several years of high numbers (Figure 3). The annual abundance of tenebrionids was not autocorrelated but tracked the annual detritus index (r^2^ = 0.80), while it did not match any other annual rain or fog related measure (r^2^ < 0.19, *p* > 0.05).

The tenebrionid community of 54 species (Table A1) showed a high Shannon diversity (H’ = 2.551) and evenness (J’ = 0.639), with 17 species accounting for 95% of the trapped individuals and the 26 focal species for 98.96%.

### 3.3. Abundance Variation

Year to year changes in the abundance of focal species (*GC* > 0.05 or <−0.05) occurred every year (Figure 4). In the gravel plain, the largest numbers of species increased in abundance during 1978, 1984, 1988, 1997, 2006–2008, 2011, 2013–2014, and 2018, all but one (1984) being in the same or the next year following effective rainfalls (Figure 2b). AV, which took both increases and decreases into account, followed different patterns over time. Species differed in terms of the time of population growth and decline, and there were few years when all species had the same trend. Abundance changes were gradual in some species and extremely variable in others (quartile 0–4 AV: 0.182, 0.525, 0.851, 1.586, 4.979; Table 1). 

Autocorrelation revealed that population fluctuations were not cyclical, except for *Zophosis devexa* and *Cauricara velox* cycling over 9–10 year (Figure 5), although only the former was significant (r = 0.33, *p* < 0.05).

All but one of the 26 focal species was recorded in at least 22 years, one in every year (Table 1). The abundance tracks of the 26 focal species could be allocated to six clusters (*p* < 0.001, Ward’s method—Pearsons-r, Figure 6). Group partners matched each other, with one species, *Cauricara eburnea* being unique, not matching any other species (*p* > 0.05, Figure 6).

*Zophosis moralesi* was the most abundant species, accounting for 20.1% of the overall total. This species belongs to group A, whose five members accounted for 40.9% of the total abundance and tracked the detritus index (r^2^ = 0.31 to 0.83). The three group B species (18.0%) were influenced immediately by effective rainfall (r^2^ = 0.32 to 0.53), with the populations irrupting in years with effective rainfall events and crashing a year later (Figure 6). Group C (14.8%) comprised eight gravel plain species that gradually increased populations over several years following heavy rainfall events, then gradually declined. The annual abundance of group C members did not correlate with any of the environmental factors considered here, as their gradual rates of increase and decrease varied. Two group D species (19.7%) tracked detritus (r^2^ = 0.10 to 0.38), with *Zophosis amabilis* consistently present every year, even across the extremely dry period of 1990–1996, when many other species were not recorded. One species, *Cauricara eburnea*, that was allocated to group E (0.4%), correlated negatively with all the other gravel plain species (r < −0.57) and with the detritus index (r = −0.36). During 1980–1990, when populations of other species were declining, this species slowly increased its numbers before gradually declining during the period following 2006 (Figure 6). Most of the seven group F species (5.3%) were first recorded during the second part of the study period (Figure 6). All members of this group were typical riverbed residents and occasionally dispersed across the plains, especially after 2006.

When examining the population attributes against other characteristics of each species (Table 1), it is notable that all abundant species (>median) were diurnal, belonging to the tribes Adesmiini or Zophosini, but there was no relationship with size (r = −0.26) or seasonality (r = −0.03). The three species with the highest AV were strictly winter active, but AV bore no relationship with any other characteristics. The only characteristics associated with particular groups were that six of the eight group C members were nocturnal or crepuscular, as were two of the three group B species, while other groups comprised diurnal species (Table 1).

During 1990–1996, abundances of all species dropped extremely low, with only 0.38% of the total captures made during this entire period. Population densities were so low that few species were recorded (Figure 4 and Figure 6), and even the otherwise abundant species of group A were no longer recorded. This drop was less severe for *Zophosis amabilis* (0.83%). Another exception was *Cauricara eburnea*, a generally uncommon species (maximum annual abundance 29). Through the 1990s, its annual abundance averaged 23% of the maximum, being the most abundant species during several of these dry years (Figure 6). Protracted periods of population lows for this species were before 1984 and after 2006. A period of extremely low density began for group C members between 1988 and 1990 through to 2002, when trapping effort was increased from 15 to 75 traps for eight months, confirming that these species were still present. After 2006, they were again recorded in the regular traps. Nearly all focal species appeared in considerable numbers after 2006, including two group B and all the group F members previously recorded seldom, or not at all, during the initial 30 years (Figure 6). 

### 3.4. Responses to Trigger Events

In years when abundance began to increase (*GC* > 0.05), monthly abundances indicated the onset of responses by focal species relative to the month of rainfall events. Sometimes, adults became active and were detected during the month or months immediately following a rainfall event. Most population responses were initiated within six months, but some species responded only 10–24 months after a trigger event (Table 1). Most population peaks occurred during the year of the event or a year after, but four peaks occurred after 3–7 years, following steady population increases after the trigger event.

Specific rainfall schedules and amounts triggered the population responses of different tenebrionid species (Table 1). The timing and magnitude of irruptions and declines differed among species and groups. Sixteen of the 26 focal species responded to heavy rainfalls, while most of the erratic range expansions of the group F members occurred in the years following heavy rainfalls. Five species responded only to winter rainfall, not summer rain, even when this was heavy. Summer rain of 11–40 mm triggered six species, including the three most abundant. Three species responded to light rainfalls, such as 8.5 mm in the winter of 1983, and some of these responses were cued seasonally, e.g., the summer active *Zophosis amabilis* increased six months following light winter rainfall events. No species responded to every rainfall event.

Short pulses of population irruption, typically lasting 1–3 years (Table 1), were interspersed with long intervals of population decline, usually 5–10 years. There were fewer years with abundance increases than decreases (Wilcoxon matched pairs: t = 60, z = 2.16, *n* = 26, *p* < 0.05). The number of irruptions balanced crashes (>200% increase and decrease, respectively), the maximum being the seven irruptions and eight crashes experienced by *Eustolopus octoseriatus* during 45 years. All but four species had *GC* values signifying irruptions, but effective irruptions and crashes could be registered erratically at generally low abundance, e.g., *Cauricara eburnea* disappearing in 1994 and reappearing a year later without responding to a trigger event (Figure 6). The sequence of heavy rainfall years 1997, 2006, 2008, 2009, 2011, 2013, and 2018 primed most species to irrupt and crash, even those that otherwise changed slowly. The most pronounced irruptions followed by crashes a year later were by *Eustolopus octoseriatus* (AV > 3.5), with the most extreme fluctuation occurring from 2010 to 2012, when its annual abundance first increased from 34 to 7909 then crashed to 30 a year later.

### 3.5. Community Composition

A Mantel test was performed on the community revealed seriation (Figure 7), i.e., the community changed successively from year to year (R_s_ = 0.224, *p* = 0.001). Exceptions were big gaps between 1996 and 1997 and, to a lesser extent, between 1995 and 1996 and 2005 and 2006, indicating profound changes in community composition (Figure 7 and Figure 8). In addition, a run of punctuated changes occurred from 1990 to 1996. A Mantel test also revealed community cyclicity, i.e., the community changed over the study period with a strong resemblance between the composition in the early and late sampling years (R_s_ = 0.418, *p* < 0.001; Figure 7). Community composition changed continuously (Figure 8), gradually, or rapidly, in phases (Figure 7). The composition departed from its initial state of the 1970s to another state in the 1980s. It then went adrift with a succession of punctuated changes from 1990 to 1996, and from 1997 to 2005 existed in another separate state. In 2006, it returned to the 1976/1977 state (Figure 7). The trajectory that followed during 2007–2019 bore some resemblance to 1977–1989. The four parts of the large 30-year cycle were reflected by populations of most species being high initially (1976–1979), then declining at various rates (1980–1989), collapsing (1990–1996), recovering at various rates (1997–2005), and returning to a high level after 2006. Species and groups tracked this cycle differently (Figure 6 and Figure 8), with *Cauricara eburnea* inverse to the others.

## 4. Discussion

### 4.1. Population Responses to Rainfall

Long term studies of insects in arid America, Australia, and the Middle East [55,56,74,75,76,77,78,79,80] report that population fluctuations and their differences among species are largely attributable to rainfall variations. The current study follows this conclusion, with the caveats that population responses are also strongly modified by rainfall timing and the life history characteristics of the component species. Patterns of population responses to precipitation were either species specific or were characteristic of clusters of species, independent from phylogeny and biogeography. Furthermore, some responses were complex, with variable lag times placing them beyond the reach of conventional statistics [56,81], requiring case by case examination of population increases relative to the occurrence of trigger events.

Rainfall, the most important trigger of Namib Desert tenebrionid population irruptions, acts on populations in various ways. First, it increases primary production, generating detritus, the basis for detritivore productivity [45]. Effective rainfall briefly stimulates ephemeral grass growth on the previously bare gravel plains (Figure 1) [23], whereas perennial plants in nearby habitats require heavy rainfall to become established and grow [17,82,83,84,85,86,87]. In the current study, total tenebrionid abundance tracked the calculated increases in detritus biomass after effective rain, and its gradual decrease. The detritus index, incorporating rapid increases and gradual decreases in resources, extends the effects of rainfall across years. The model needs empirical confirmation and refinement, especially in terms of the decay of this reserve, adjusted for different habitats. Nevertheless, the principle of using such a model was vindicated by the outcome. Conversely, it can be argued that the total tenebrionid population should be an indicator of the relative availability of detritus over time, which can serve to guide investigations of detritus stocks and their decomposition, to improve modelling.

Effective rainfall in summer, especially when heavy, appears to be the principal driver of the populations of the dominant perennially active species (group A), as well as of the ephemerally active tenebrionid species (group B) and uncommon beetles (group C). The latter only attained sufficient numbers to be recorded in years following heavy rains, gradually increasing in abundance until they could be the most abundant species in the community eight years later (Figure 8). These three groups constituted more than three-quarters of tenebrionid numbers, explaining the aforementioned relationship to the detritus index.

Rainfall also increases soil moisture for several weeks [43,44,88], favouring reproductive success and recruitment [52,89]. For the group B species, with quick and short responses to rainfall, the population peak occurred while shallow soil moisture was suitable for larval development and there was green grass for consumption [26]. In winter, even relatively light rainfall triggered responses by several species (groups D–E, Table 1), probably because soil moisture lasted longer in winter due to reduced evaporation from 3–9 °C cooler soil [43,60].

Rainfall in early winter (April and May) was a principal driver of the group D populations. *Zophosis amabilis* was active throughout the year, peaking in summer. This species, however, showed no strong population responses to summer rains, even if heavy. Its adult population increased six months after winter rainfalls of >5 mm. *Cauricara velox* was strictly winter active, with its population responding only to effective winter rains in either the same season or a year later. Besides these species, *Zophosis cerea* and *Rhammatodes tagenesthoides* of group C and *Cauricara eburnea* of group E were also cued to winter rains. Furthermore, responses to light winter rains were prevalent even with other species, e.g., in 1984, eight species increased in abundance a year after 8.5 mm winter rain in 1983.

The mechanisms by which winter rainfall drives populations are unknown. In the Central Namib, rains falling in April and May (Figure A1) are normally the last of the summer monsoons (therefore, conventionally considered part of a summer rainfall regime), but the resultant lingering moisture in the cool months of early winter probably resembles the effects of the winter rainfall fronts of the southern Namib and Western Cape [90,91]. The lingering moisture would also affect plants, perhaps explaining the conflicting interpretations concerning winter and summer rainfall zones in the Namib [92]. In arid areas elsewhere, it has been noted that the significance of soil moisture changes with seasons, facilitating the enhanced growth of some plants during cool months [91]. Different species partition their access to water over time [93]. We could be seeing a similar process with the tenebrionids, explaining the diverse population patterns between response groups and, to a lesser extent, within groups. The current recognition of high species diversity enabled by complex time niches elucidate Barrows’ [94] hypothesis that relationships to hydrological patterns could explain the high diversity of tenebrionids he found in the Coachella Valley dune field in California.

### 4.2. Population Responses to Fog

The increase in Namib fog from low, in the 1970s, to high, during the 1990s (Figure 2c), allowed the separation of the effects of fog and rainfall. Particularly during the high fog years of the early 1990s, when all indices related to rainfall were low and most species’ populations were declining, associations of beetle populations with fog were implicated for the eight Namib species that declined more gradually than other species [49]. Fog directly benefits tenebrionids [39,95,96,97], but it also enables some perennial plants that take up fog water [98,99,100,101,102] to constantly produce seeds and plant fragments, supplementing food for detritivore populations.

The current study revealed that *Cauricara eburnea*, which is not known to drink fog [49], increased slightly through the 1990s, in an opposite trend to all the other species. *Cauricara eburnea* is more commonly encountered in lichen fields closer to the coast (Figure 9), where fog is more prevalent [103]. When fog was prevented from reaching the larvae of the eight tenebrionid species at Gobabeb, they either died (three species) or developed more slowly [52]. Perhaps *C. eburnea* benefits from fog due to enhanced larval development or the improved palatability of fog wetted food. The absorption of moisture by detritus during humid conditions [40] enhances its fungal community [21], a potential food source for tenebrionids.

### 4.3. Recovering from Low Abundance

Many periodically abundant species were sometimes scarce for extended periods, joining other perpetually rare species occurring at the study site. The recorded annual fluctuation in the number of focal species present in any given year (Figure 4) is probably an illusion of the local extinction of some species. A temporary fivefold increase in trapping effort during the abundance trough revealed that previously unrecorded species were still present at the site, though at very low densities. In contrast to mesic areas, where tenebrionid species dropping from common to rare is considered a sign of an increased risk of extinction [104], this study indicates that many Namib species cope with temporary rarity. One of the key topics of biodiversity is how rare species persist [105], whether in a disturbed tropical forest [106] or a hyperarid environment subject to extreme pulse–reserve fluctuations.

Diapause has not been demonstrated in Namib tenebrionids. Eggs are not dormant. Larvae can prolong their development by several months [52,89]. However, even the longest larval development, of 19 months, would usually not be sufficient to carry a population through from one effective rainfall to the next. By contrast, adults of all the eight species examined by Rössl [52] had a minimum natural lifespan of 27 months and up to 73 months (or longer, observations were terminated). These observations indicate that adults are, by far, the most enduring life stage, lasting several years, and individuals may experience more than one effective rainfall. Extreme longevity by adults is, thus, a key feature of Namib tenebrionids. Fog, dew, nocturnal subsurface vapour [38,107], and light rain in winter appear important for sustaining them long after effective rainfalls [39].

Immigration could be a mechanism of recovery. Local populations could be replenished from a neighbouring source population. Although tenebrionids are apterous, they have been observed to move several kilometres during their lifetime [95,108,109,110]. Such species respond to triggering events by increasing in abundance in their main habitat and expanding their range. For example, the seven group F species are usually encountered in the riverbed of the Kuiseb River [59]. After a series of unusually high rainfalls and river flooding events between 2006 and 2018, these riparian species ventured more than 2 km across the grass covered gravel plain, where they were recorded.

Repopulation from distant locations may be slow. The maximum ranging distance of the most mobile species, *Onymacris plana*, is 10 km (R. Pietruszka personal communication), barely far enough to cover the path width of a thundercloud (Figure 1), and far less than the much greater distances (tens to hundreds of kilometres) between adjacent annual rainfall paths across the Namib [22,24]. In addition, there may be times when no rain falls for many years over an entire region of the Namib, and all populations of a species become scarce. Such populations can only recover by a few survivors reproducing.

The ability for rapid local recruitment by resident populations is, therefore, a key feature. The timing of rainfall strongly influences the timing and magnitude of responses. Summer rain allows summer active adults to multiply immediately. For instance, *Z. moralesi* reproduced, developed, and metamorphosed in 75 days or less [52]. It is, therefore, possible to recruit a fresh cohort of adults within the same annual breeding season and increase the irruption. 

Namib tenebrionid populations have several characteristics consistent with the “storage effect” [111], which facilitates coexistence among species that rely on common resources [112]. Existing in a fluctuating environment, Namib beetles regularly pass through bottlenecks of low resource availability when reduced to a few individuals. These individuals are mature, capable of reproducing at any time, are long lived, and have overlapping generations. They are the population storage modules upon which regeneration is based when a pulse enables populations to increase again. When the survival of developing offspring in unpredictable environments is uncertain, bet-hedging reproduction is a successful strategy [113]. Iteroparous females of many Namib tenebrionids frequently produce small clutches of eggs [45,46,52,57,89,114,115,116,117,118]. Females forage until they have enough resources to produce a clutch, lay it, and then forage again to produce the next clutch. Under optimal conditions, a female can produce one or more clutches daily, and some species do so throughout the year. 

The latency of response in terms of tenebrionid population increase after rainfall depends on the time it takes for germinated ephemeral grass to turn into detritus, approximately one month, plus the shortest interval required for the next cohort of tenebrionids to develop. For Zophosini, this is 2–3 months (i.e., 3–4 months after rainfall), and for Adesmiini, 5–7 months (i.e., 6–8 months after rainfall) [52]. These periods roughly correspond with the lags for sustained increases in abundances (Table 1). However, not all species followed the “detritus” pattern, and those which did, did so at variable rates. Even within the numerically dominant group A, where annual abundance patterns were most strongly correlated (r = 0.47–0.80), patterns differed in detail (Figure 6 and Figure 8), while members of other groups differed starkly, even those cued to common triggers. The strictly winter active *C. velox* sometimes only irrupted a year after winter rainfalls, indicative of lags in their recruitment capacity when triggered. Population responses to the triggers of some species operated within the constraints of differing phenology and larval longevity. 

Differences in responses to pulses and rates of population decline result in time partitioning within the community. This partitioning could explain how so many Namib tenebrionid species coexist. Previous studies have recorded 41 tenebrionid species at the current gravel plain study site. With continuous trapping, the number could be as high as 54 (not all current records have been confirmed), with 82 tenebrionid species recorded in several habitats within a kilometre of Gobabeb [59]. Although the extraordinary high diversity and species radiation of Namib tenebrionids have been further illuminated since Koch’s [28] landmark publications on that subject [66,119,120,121,122,123], several hypotheses remain untested. The current study explains how different relationships to water and food availability enable so many species to persist sympatrically.

### 4.4. Population Variability

Wolda [70] summarised a widespread search for correlates and principles concerning the variability of insect populations, evaluating the relationship between stability and biome derivation. He considered the significance of mean AV values for 138 sets of temperate and tropical insects [70]. Only one of Wolda’s AV values was higher than the highest value in the current study, three Namib species lie below Wolda’s median, and only *Epiphysa arenicola* falls within Wolda’s lower quartile. The mean AV value of all the focal species of Namib tenebrionids in the gravel plain (0.858) lies above Wolda’s upper quartile of values (quartile 0–4: 0.018, 0.206, 0.412, 0.666, 5.250). It should, however, be noted that the minimum annual abundance of all but one of the current focal species was zero, which tends to render AV values high [70].

What is the significance of the wide range of AV values for tenebrionid species (0.182–4.979, Table 1)? An answer to this question requires a species by species analysis of the correlates of similarities and differences in the AV values. The species with the highest AV, *Eustolopus octoseriatus*, is known to feed on fresh grass after effective rain [26], in other years remaining quiescent, buried in the sand for several years until the next rain. AV values > 1 of some of the other species were due to the periodic local range expansions, from the riverbed onto the gravel plain, by seven species during the decade of high rainfalls following 2006. The lowest AV value (0.182) was for *E. arenicola*, one of the few nocturnal focal species that only responded to the successive exceptionally heavy rainfalls of 1976/1978 and 2006/2011, with the first increase in adults showing two years after the last of these event pairs, peaking three years later and gradually declining over the next decade (Figure 6). Another low AV (0.288) was for *Z. amabilis*, a species recorded every year, the most responsive of all the species to any winter rainfall, even if only light. *Cauricara velox*, a highly seasonal winter active species with low AV, is a small, long legged species that is highly mobile, searching large areas of gravel plain for widely scattered detritus, of which it continued to find sufficient to moderate population change. These examples illustrate how the wide AV range reflects sympatric tenebrionids existing in different time niches.

### 4.5. Long Term Observations

This study emphasises the fundamental rationale of long term ecological research [124]. Forty five years of observations enabled the recording of some species that were scarce for long intervals and common for short periods, and allowed the identification of population characteristics and novel analyses of tenebrionid population dynamics in the Namib Desert. Identification of the triggers to population increases required replicated observation of sporadic rainfall events. Identifying species with similar population signatures required long datasets across extended periods with different hydrological patterns.

Critical for achieving the continuity required for obtaining such long term data is the continuous institutional operation of field stations, in this case, the Gobabeb Namib Research Institute, founded in 1963 to improve understanding of the Namib Desert compared to other arid and mesic systems globally [125]. Labour intensive live pitfall trapping, requiring over 150 person-days annually, year-in-year-out, continuously conducted at three sites and intermittently at three others [59], was possible only in the close vicinity of the field station with the participation of interns as part of their training in ecology.

Long term studies of invertebrates and their climatic and ecological drivers in arid environments are rare [81]. The current analyses enhance the value of continuing the long term observations of Namib tenebrionids. There is a wealth of further information in the existing published dataset [69]. Further analyses could include examining the biological mechanisms of extreme irruptions (e.g., *E. octoseriatus*) or the ecological underpinnings of responses to winter rainfall. The pervasiveness of rarity in the desert, whether a temporary bottleneck or a permanent characteristic of species, can best be investigated against the background of long term records. While there have been numerous studies of the common tenebrionid species in the Namib, especially Adesmiini, the current long term records encourage ecological and ecophysiological studies of other species, in order to improve understanding of the extraordinary tenebrionid diversity of the Namib. Continuous records of the kind assembled here provide an invaluable foundation for short term studies to build an interconnected picture of ecosystem functioning that other means cannot achieve.

Climate change is a core feature of LTER. Specifically, in this context, there are concerns worldwide about insect populations collapsing due to climate change [126,127]. While the current 45-year dataset lends itself to such analyses, this requires examining several other factors not considered in the current paper. The records of the Namibia Meteorological Service indicate that, in the 140 years before 2020, three of the four years when rainfall exceeded 100 mm in the hyperarid Central Namib occurred during the 45 years of this study. The unusual sequence of high rainfall years (>40 mm) after 2006 was accompanied by surges in the abundance of many species and a return to a previous condition, not only of tenebrionids and ephemeral grasses but also perennial plants [17,82,83,84,85,86,87]. This long term resilience of Namib communities could reflect the high tenacity of desert organisms and/or general ecosystem health in a large national park with minimum pollution and none of the blanket application of pesticides commonly practised across rangelands and croplands [128]. Further analyses need to incorporate temperature and carbon sequestration or emission and microbial ecology to enable a more in depth understanding of Namib ecology relating to tenebrionid populations faced with climate change.

## 5. Conclusions

Detritivorous tenebrionids persist in the hyper-arid Namib because detritus produced by rainfall events prolongs the presence of their populations. Adults are long lived and maintain a bet-hedging reproductive strategy, and many use NRM to overcome water limitations during long intervals without effective rainfall events, allowing them to irrupt after the next rainfall pulse.

Many species respond to particular kinds of rainfall—e.g., summer rainfall with its flush of primary productivity, or winter rain with its lingering soil moisture—or require different intensities of such events to respond at all. Despite the wide range of triggers, initial response times, and population change rates, species show several discrete nodes, defining the common population signatures of abundance patterns. Population signatures show categorical differences among species in how they respond to rainfall of different magnitudes in different seasons. Nevertheless, nearly all population changes can be related to hydrological pulses. 

These observations of the fundamental importance of hydrological triggering events concur with Noy-Meir’s [129] autecological perspective. The population changes of individual species in the study area are driven by hydrological events but are species specific, multifaceted, and affected by several environmental or antecedent conditions besides rainfall. By responding selectively to alternative triggering events, sympatric species show temporal patterns and magnitudes of population fluctuation more diverse than predicted by Noy-Meir’s [3] original model. The great age of the Namib [130] would allow sufficient time for these divergent patterns to evolve.

## Figures and Tables

**Figure 1 insects-12-00804-f001:**
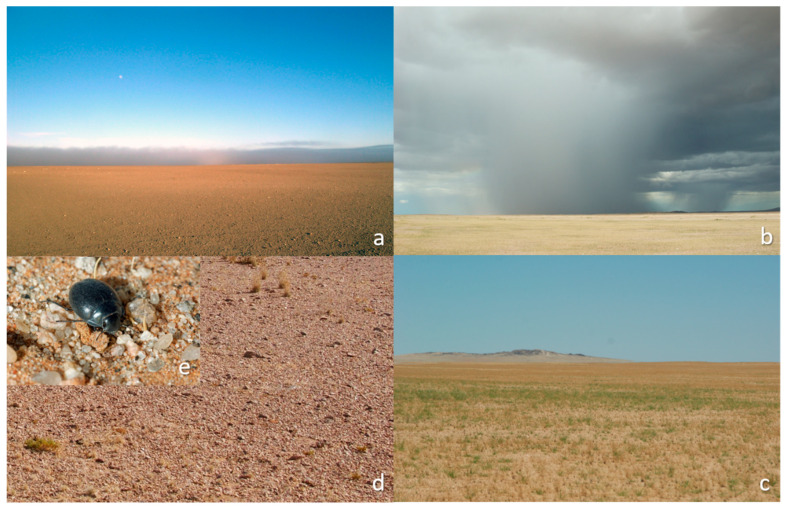
Clockwise from top left: (**a**) bare gravel plain with a distant fog cloud during an interpulse year, (**b**) effective rainfall driving the production of ephemeral grasses, (**c**) which dries, (**d**) becomes detritus, (**e**) and is consumed by tenebrionid beetles such as *Zophosis moralesi*.

**Figure 2 insects-12-00804-f002:**
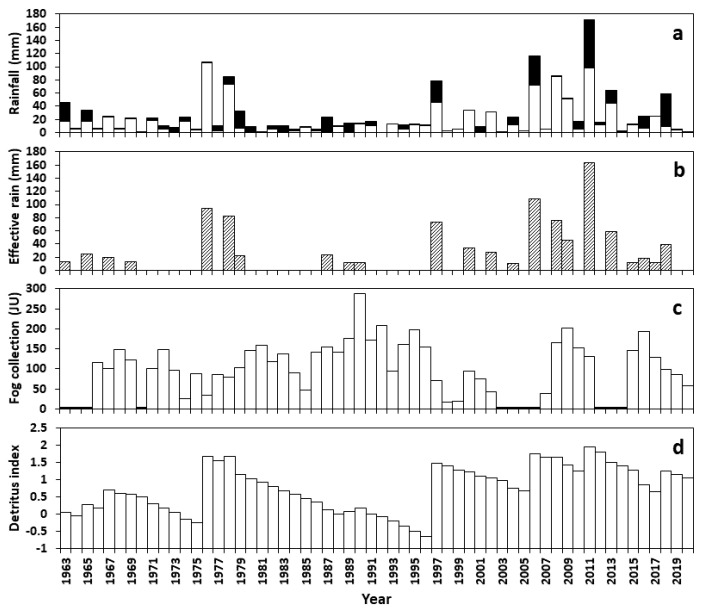
(**a**) Annual rainfall (mm) recorded at Gobabeb between 1963–2020, separated into summer rainfall (white) and winter (black). (**b**) Effective rainfall events (mm). (**c**) Annual fog precipitation (Juvik units); black bars denote missing data. (**d**) The annual index of detritus quantity calculated from grass productivity and decomposition rates affected by moisture in the central Namib.

**Figure 3 insects-12-00804-f003:**
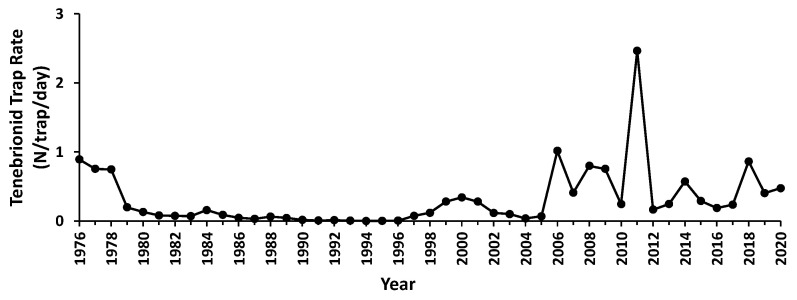
Time series of the annual trapping rate of tenebrionids (N trap^−1^ day^−1^) at the gravel plain study site.

**Figure 4 insects-12-00804-f004:**
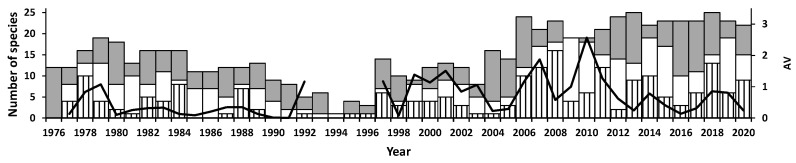
Number of focal species recorded increasing (hatched), decreasing (white), or maintaining (grey) abundance in different study years. The line graph shows annual variation (AV).

**Figure 5 insects-12-00804-f005:**
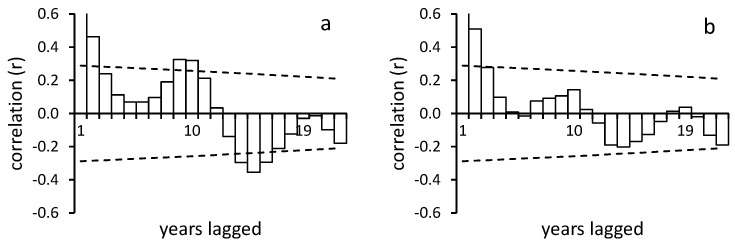
Autocorrelations of the annual abundances of (**a**) *Zophosis devexa* and (**b**) *Caurica velox*. Dotted lines are confidence limits.

**Figure 6 insects-12-00804-f006:**
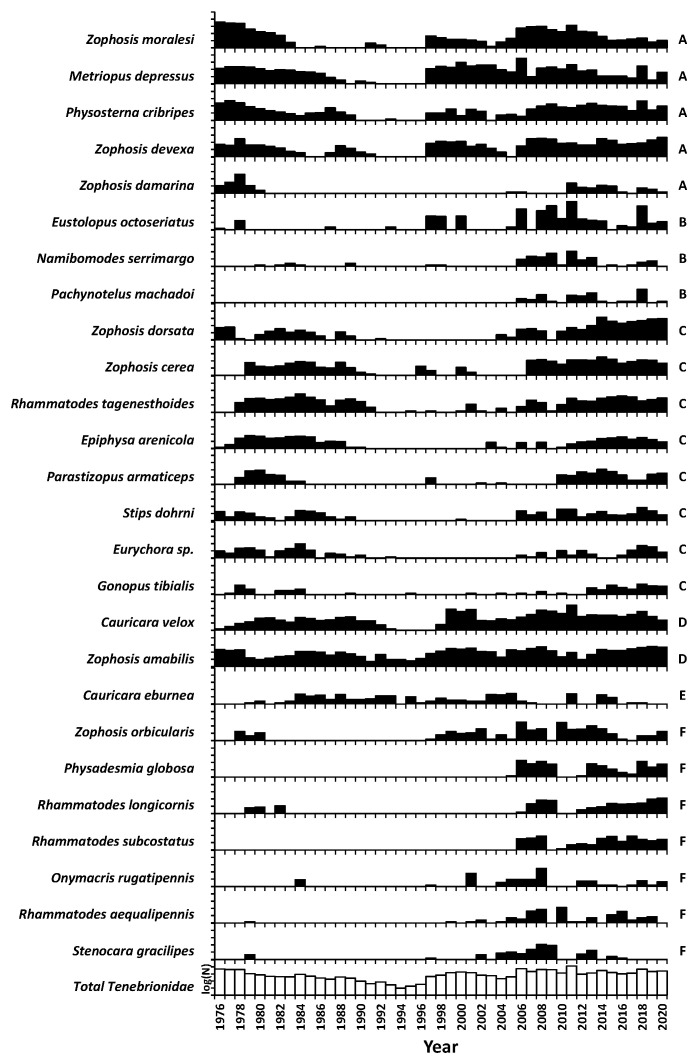
Time series of the standardised annual abundance of focal species on the gravel plain. Members of groups A–F correlate (r = 0.47–0.89, *p* < 0.001) with other group members and are linked with cluster analysis.

**Figure 7 insects-12-00804-f007:**
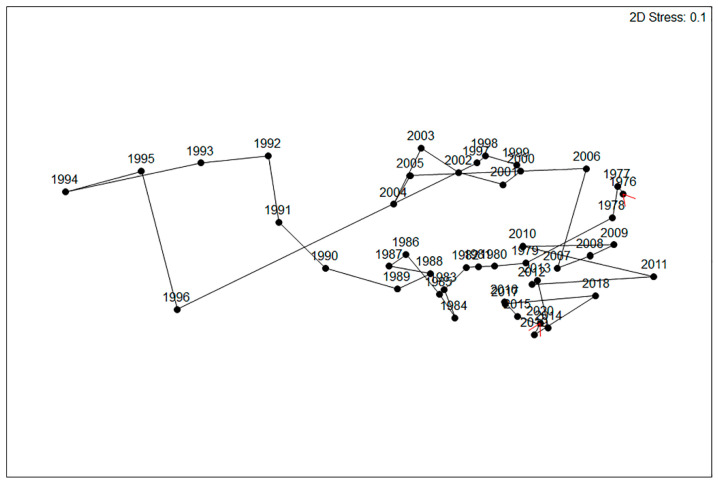
Nonmetric multidimensional scaling ordination based on Bray–Curtis similarities of focal species in different years at the gravel plain study site. Red arrows mark the start and end years of the study. Consecutive years are connected.

**Figure 8 insects-12-00804-f008:**
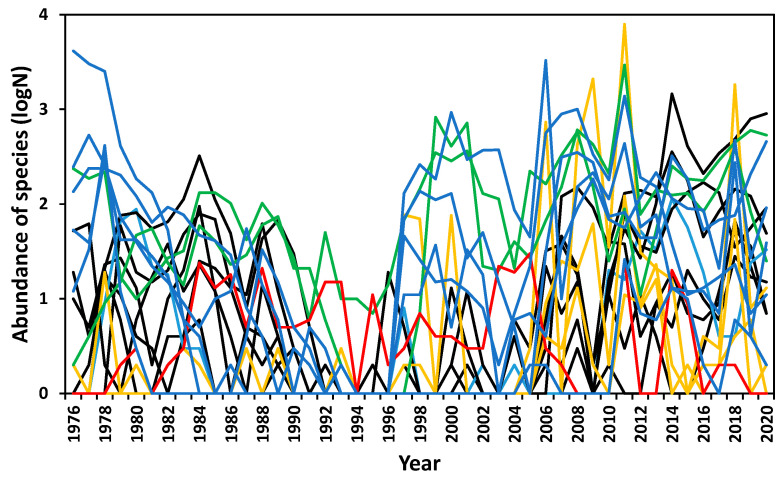
Annual changes in community composition at the gravel plain study site showing members of group A (blue), B (yellow), C (black), D (green), and E (red). Group F members, which appeared mainly after 2006, are not shown.

**Figure 9 insects-12-00804-f009:**
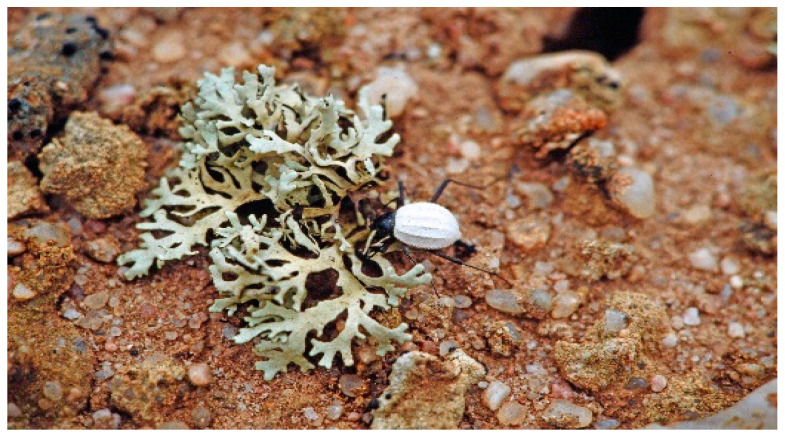
*Cauricara eburnea* feeding on lichen in the fog zone northwest of Gobabeb.

**Table 1 insects-12-00804-t001:** Abundance measures of focal species [69]: average and maximum trap rate per 1000 trap days, the number of years in which species were recorded and populations increased (*GC* > 0.05) or decreased (*GC* < −0.05), annual variation (AV), the number of population irruptions (*GC* > 1.3) recorded over the study period. The next set of columns describes which kind of rainfall triggered population increases (S = effective rainfall in summer, Oct–Mar; W = effective rainfall in winter, April–September; H = heavy rainfall, >40 mm; L = light rainfall, <11 mm), the number of months for the population to show first consistent response, the number of years to reach a peak and the number of years until the end of a response. The right columns are characteristics of the species: dry body mass (mg) [73], diel activity (D = diurnal, N = nocturnal, C = crepuscular) [64,68], and the principal season of activity according to winter records (<33% = summer (S), >66% = winter (W), 33–66 = aseasonal (A)).

Gravel Plain Species	Tribe	Group	Capture Rate/1000 Traps	Maximum Capture Rate	Years Recorded	Years Increasing	Years Decreasing	Annual Variability (AV)	Times Irrupted (*GC* > 1.3)	Rainfall Trigger	Month First Response	Year Peak Response	Duration of Response	Mass (mg)	Diel Activity	Winter Activity (%)	Principal Season
*Zophosis (Gyrosis) moralesi* (Koch)	Zophosini	A	57.59	753	35	6	16	0.516	2	S, H	1–19	1–3	3–4	33	D	30	S
*Metriopus depressus* (Haag)	Adesmiini	A	35.41	601	39	13	16	1.012	3	S, H	1–10	1–2	4–10	54	D	65	A
*Zophosis (Gyrosis) devexa* Peringuey	Zophosini	A	13.27	83	37	13	16	0.525	2	S, H	1–8	1–2	3–7	12	D	38	A
*Physosterna cribripes* (Haag)	Adesmiini	A	11.31	97	38	12	14	0.534	3	S, H	2–8	1–4	3–11	402	D	48	A
*Zophosis (Occidentophosis) damarina* Peringuey	Zophosini	A	2.36	76	16	3	2	1.719	1	H	1–24	1	1	32	D	60	A
*Eustolopus octoseriatus* Gebien	Adesmiini	B	44.04	1445	21	8	9	4.979	7	S, H	1–4	1	1–2	105	D	94	W
*Namibomodes serrimargo* (Gebien)	Molurini	B	0.86	22	19	3	3	2.004	1	H	2–3	1	3–4	33	N	91	W
*Pachynotelus machadoi* Koch	Cryptochilini	B	0.57	12	12	4	3	1.994	2	H	3–4	1	1–2	71	C	86	W
*Zophosis (Z.) dorsata* Peringuey	Zophosini	C	18.85	266	29	15	13	0.641	2	H	1–7	1–2	3–4	56	D	23	S
*Zophosis (Occidentophosis) cerea* Peringuey	Zophosini	C	8.58	65	31	12	13	0.713	4	W	6–18	2–3	3–5	27	D	32	S
*Rhammatodes tagenesthoides* Koch	Tentyriini	C	8.31	59	35	16	10	0.422	1	W	3–13	3	3–5	3	N	48	A
*Epiphysa arenicola* Penrith	Adesmiini	C	2.51	11	30	6	7	0.182	0	H	23–24	4–5	10–15	384	C	30	S
*Parastizopus armaticeps* (Peringuey)	Opatrini	C	2.29	21	21	7	6	0.583	1	H	3–29	3–4	7–8	160	N	34	A
*Stips dohrni* (Haag)	Eurychorini	C	1.45	12	29	6	7	0.851	2	H	1–6	1–2	2–5	47	N	72	W
*Eurychora sp. A*	Eurychorini	C	1.50	17	27	5	6	0.741	1	H	1–17	2–7	3–10	70	C	60	A
*Gonopus tibialis* Fabricius	Platynotini	C	0.62	5	21	3	3	0.455	0	H	24–25	3	7–8	321	N	63	A
*Cauricara velox* (Peringuey)	Adesmiini	D	30.20	538	41	14	16	0.497	1	W, H, L	1–20	1–3	3–5	35	D	96	W
*Zophosis (Calosis) amabilis* (Deyrolle)	Zophosini	D	27.51	109	45	18	18	0.288	0	W, L	1–10	1–3	3–4	36	D	24	S
*Cauricara eburnea* (Pascoe)	Adesmiini	E	1.13	5	32	7	7	1.274	3	W, L	6–18	1–2	1–4	44	D	57	A
*Zophosis (Gyrosis) orbicularis* Deyrolle	Zophosini	F	3.85	67	25	8	8	1.879	3	S, H	11–24	1–3	3–4	62	D	39	A
*Physadesmia globosa* (Haag)	Adesmiini	F	3.20	35	14	5	6	1.586	3	?				236	D	67	W
*Rhammatodes longicornis* Haag	Tentyriini	F	1.71	24	16	6	2	1.457	1	?					N	65	A
*Rhammatodes subcostatus* Koch	Tentyriini	F	1.98	61	14	7	4	1.049	1	?				7	N	47	A
*Onymacris rugatipennis* (Haag)	Adesmiini	F	1.93	61	16	2	2		2	?				235	D	74	W
*Rhammatodes aequalipennis* Peringuey	Tentyriini	F	1.04	27	18	5	3	2.830	2	?					N	58	A
*Stenocara gracilipes* Solier	Adesmiini	F	0.87	22	13	3	3	1.561	0	?				110	D	70	W

## Data Availability

The dataset and metadata are published at http://data.sasscal.org/metadata/view.php?view=ts_timeseries&id=7235&ident=416501860743123172 (accessed on 15 August 2021) [69].

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
