# Peer review of "Long-Term Population Dynamics of Namib Desert Tenebrionid Beetles Reveal Complex Relationships to Pulse-Reserve Conditions"

_insects, 2021, doi:10.3390/insects12090804_

Round 1

Reviewer 1 Report

Congratulations on the very interesting and well-done study. Besides the long time span of field research, I really acknowledge how the data are presented and that the author describes new findings on which he builds well-supported conclusions.

I have several small proposals:

Abstract There is no information on the number of species/numerical dominance of any pattern (patterns a-f) 

line 126 Please, consider the structure of the sentence: 

...data were collected from 1963-2020 at ...

line 215 Please, consider the expression: 

p-values were 0.05       (were they exactly 0.05, I expect lower than 0.05)

lines 289-300

A bit inconsistent presentation, present separately species and/or groups or eventually add info on species included in each group. Sometimes,  you start with the most common species of the group, sometimes, you do not mention any.

Author Response

Thank you for your words of appreciation of the manuscript. My responses to your specific suggestions (bullet-points) are in the sub-bullets:

  • Abstract There is no information on the number of species/numerical dominance of any pattern (patterns a-f) 
    • this information was added for each pattern, which now reads: "Six patterns of population variation were recognised: a) increases triggered by effective summer rainfalls, tracking detritus over time (5_species, 41% abundance); b) irrupting upon summer rainfalls, crashing a year later (3, 18%); c) increasing gradually after series of heavy (>40 mm) rainfall years, declining over the next decade (8, 15%); d) triggered by winter rainfall, population fluctuating moderately (2, 20%); e) increasing during dry years, declining during wet (1, 0.4%); f) erratic range expansions following heavy rain (7, 5%)."
  • line 126 Please, consider the structure of the sentence: ...data were collected from 1963-2020 at ...
    • the sentence was restructured: "From 1963-2020, rainfall data were collected at the Gobabeb First Order Weather Station."
  • line 215 Please, consider the expression: p-values were 0.05 (were they exactly 0.05, I expect lower than 0.05)
    • revised: "p-values were less than 0.05"
  • lines 289-300: A bit inconsistent presentation, present separately species and/or groups or eventually add info on species included in each group. Sometimes,  you start with the most common species of the group, sometimes, you do not mention any.

    • the revised paragraph provides consistent information. The three species that are mentioned by species name are because these were most abundant of all, or always present, or showing opposite trends to others. The revised text now reads: "Zophosis moralesi was the most abundant species, accounting for 20.1% of the overall total. This species belongs to group A, whose five members accounted for 40.9% of the total abundance and tracked the detritus index (r2=0.31 to 0.83). The three group B species (18.0%) were influenced immediately by effective rainfall (r2=0.32 to 0.53), with the populations irrupting in years with effective rainfall events and crashing a year later (Figure 6). Group C (14.8%) comprised eight gravel plain species that gradually increased populations over several years following heavy rainfall events, then gradually declined. The annual abundance of group C members did not correlate with any of the environmental factors considered here, as their gradual rates of increase and decrease varied. Two group D species (19.7%) tracked detritus (r2=0.10 to 0.38), with Zophosis amabilis consistently present every year, even across the extremely dry period 1990-1996 when many other species were not recorded. One species, Cauricara eburnea, that was allocated to group E (0.4%) correlated negatively with all other gravel plain species (r<-0.57) and with the detritus index (r<-0.33). During 1980-1990 when populations of other species were declining, this species slowly increased its numbers before gradually declining during the period following 2006 (Figure 6). Most of the seven group F species (5.3%) were first recorded during the second part of the study period (Figure 6). All members of this group were typical riverbed residents and occasionally dispersed across the plains, especially after 2006."

Reviewer 2 Report

In this study, the author used a long-term (1976-2020) data series about tenebrionid beetles in the Namib desert to test the hypothesis that desert populations respond rapidly to occasional rainfall events and maintain overall stability. This is a very fine manuscript, well written, and it is also a perfect fit for the selected special issue. I only have minor comments that I hope could improve the manuscript further.

  1. Perhaps an explanation about how the tenebrionid groups were categorised should already appear in the M&M section, otherwise, it comes a bit as a surprise.
  2. Figure 8 is virtually unreadable and, in this form, does not add much to the text. I would either remove it or change it in a dot chart. I believe that a dot chart version would be much easier to read. 
  3. Throughout the text, the UK and US spell have been mixed (e.g., L179 standardized & analyse). It would be more elegant to use only one of them. 
  4. L111 perhaps this sentence would be more appropriate after the main text in the acknowledgement. 
  5. L262 perhaps better “the tenebrionid  community showed a high Shannon diversity….”
  6. In the reference list, there are a few scientific names not in italics and inconsistency. I highlighted the ones I spotted in the PDF version.

Author Response

Thank you for the positive reflection on the manuscript. My responses to your minor comments (numbered points) follow as sub-points.

  1. Perhaps an explanation about how the tenebrionid groups were categorised should already appear in the M&M section, otherwise, it comes a bit as a surprise.
    • This is now explained in the M&M methods, where abundance correlations are described (L188 following): "Abundance fluctuations were tested for cyclicity by autocorrelation (lagged 3-22) of log(N+½) and GC. Finally, species were compared by correlation (Pearson’s r), which formed the basis of distance measures of a cluster analysis, using Ward’s method of hierarchical grouping attained by minimising Sum of Squares [71]. Species that correlated and clustered most closely (P<0.001) were assigned to distinct groups." Ward (1963) was added to the reference list.
  2. Figure 8 is virtually unreadable and, in this form, does not add much to the text. I would either remove it or change it in a dot chart. I believe that a dot chart version would be much easier to read. 
    • Use of the stacked histogram was inspired by its use in Fig.1 of the following publication: Naidoo, Y.;  Valverde, A.;  Pierneef, R. E.; Cowan, D. A., Differences in Precipitation Regime Shape Microbial Community Composition and Functional Potential in Namib Desert Soils. Microb Ecol 2021, DOI: 10.1007/s00248-021-01785-w. I could not get the desired effect with a dot-chart. I modified Figure 8 by overlaying multiple line graphs colour-coded according to group membership. This figure illustrates an important characteristic underlying Figure 7, namely that community composition changed constantly as the trajectories differed between species and groups. The new Figure 8 may perhaps not be easier to read than the previous version if trying to follow species by species, year by year, but the main message of constant change should be clear from the overall pattern. For that reason, a species legend was omitted.
    • In the text, I added another citation to Figure 8 in L426: "Community composition changed continuously (Figure 8), gradually or rapidly in phases (Figure 7)."
    • In L434: "Species and groups tracked this cycle differently (Figure 6 & 8), with Cauricara eburnea inverse to others."
    • The figure caption L446 now reads: "Figure 8. Annual changes in community composition at the gravel plain study site showing members of group A (blue), B (yellow), C (black), D (green), and E (red). Group F members, which appeared mainly after 2006, are not shown."
    • In the discussion, Figure 8 is now referred to in L484 to explain the changeable nature of community composition: "and uncommon beetles (group C). The latter only attained sufficient numbers to be recorded in years following heavy rains, gradually increasing abundance until they could be the most abundant species in the community eight years later (Figure 8). "
  3. Throughout the text, the UK and US spell have been mixed (e.g., L179 standardized & analyse). It would be more elegant to use only one of them. 
    • UK spelling is used and US spelling changed throughout
  4. L111 perhaps this sentence would be more appropriate after the main text in the acknowledgement. 
    • The dedication now stands at the end of the acknowledgements
  5. L262 perhaps better “the tenebrionid  community showed a high Shannon diversity….”
    • The sentence now reads: The tenebrionid community of 54 species (Table A1) showed a high Shannon diversity (H’=2.551) and evenness (J’= 0.639), with 17 species accounting for 95% of the trapped individuals and the 26 focal species for 98.96%.

  6. In the reference list, there are a few scientific names not in italics and inconsistency. I highlighted the ones I spotted in the PDF version.
    • I did not get to see the highlights in the PDF version, but I combed through the reference list, italicised species names and brought consistency into the capitalisation of reference titles. These changes were implemented but not shown in Track Changes.

Reviewer 3 Report

This is an excellent manuscript, demonstrating the value of long-term data sets for providing a fine-scale assessment of the drivers of species abundance and richness in a hyper-arid desert environment. The Namib Desert gravel plains lack spatial diversity, but over time there is considerable temporal diversity in the amount of available moisture. It is the temporal diversity that has created different niches and has allowed some 26 species of tenebrionid beetles to become established there. Any desert ecologist knows that moisture drives the abundance of species, but here we were able to see that the patterns of moisture availability can also drive the number of species as well.

I have little to criticize, however, the one thing the author could have included was more discussion of how species richness was driven by diverse rainfall patterns. Long ago I published on tenebrionid beetle species richness in a sand dune system in North America, but there I found spatial diversity in dune characteristics along with spatial moisture gradient to explain the 30 or so species I found there. So finding similar levels of species richness with only temporal diversity is extraordinary. 

Author Response

Thank you for your positive reflection. My responses to your suggestion (bullet) follows in sub-bullets.

  • the one thing the author could have included was more discussion of how species richness was driven by diverse rainfall patterns. Long ago I published on tenebrionid beetle species richness in a sand dune system in North America, but there I found spatial diversity in dune characteristics along with spatial moisture gradient to explain the 30 or so species I found there. So finding similar levels of species richness with only temporal diversity is extraordinary. 
    • I inserted minor modifications in the text concerning Namib species responses to rain, and at L512 added the sentence:  "The current recognition of high species diversity enabled by complex time niches, elucidate Barrows’ [94] hypothesis that relationships to hydrological patterns could explain the high diversity of tenebrionids he found in the Coachella Valley dune field in California."

    • in L614: "The current study explains how different relationships to water and food availability enable so many species to persist sympatrically."
    • in L642:  "These examples illustrate how the wide AV range reflects sympatric tenebrionids existing in different time niches."

Reviewer 4 Report

This is a very fine study, impressive both for the extraordinarily long term and field-intensive nature of the dataset (45 years), and for the depth of the insights that have been extracted from it. By spanning such a long period, population fluctuations of many tenebrionid beetle species have been able to be compared with the rainfall, fog and consequent resource conditions that prevailed through the study, allowing six different patterns of population variation to be discerned in the beetles. This diversity of responses introduces nuance into our understanding of the dominant pulse-reserve paradigm of Noy-Meir and, importantly, provides a novel mechanism to explain the very high diversity of tenebrionids in the Namib Desert system. The long-term data have been analysed competently and appropriately, and the manuscript is clear and well written. (I noticed only one typographical error: at line 23 'explains' should be 'explain'). In consequence, I can offer only a small number of minor suggestions to strengthen this very impressive work.

Lines 50-53: In stating that "Relatively few studies have been concerned with animal populations that bridge interpulse periods while maintaining the capacity to irrupt in response to sudden increases in water and food availability," I suggest this is correct for invertebrates but not for all desert animals. There have been, for example, quite a number of long-term studies of birds, mammals, frogs and reptiles in global desert environments, with many of these encompassing the reserve phase and showing marked population irruptions after heavy rainfall. Perhaps some of these studies could be cited, or the sentence reworded to reflect that the statement refers primarily to invertebrates.

Lines 169-171: Pitfall traps were monitored twice or thrice weekly and beetles released near the site of capture. Thus, beetles could be in traps for 2-3 days before being checked. Is there any chance that the recorded beetle numbers could have been affected by predators visiting the traps and removing beetles between checks? If predation was a constant, this would not be an issue. But if predators such as mobile birds or mammals increased in response to the pulses of rainfall, could these have temporarily increased predation pressure on the trapped beetles and introduced bias in the numbers that were counted in the traps? I suspect that any such biases may wash out over 45 years, but the author may wish to discuss whether predators were present and could have had potential effects.

Line 218: Here and later in the results, please clarify what ± refers to. I presume it is standard deviation, but it is worth specifying.  

Line 251: I suggest rewording "A total of 63,291 tenebrionids were captured" to something like "In total, 63,291 captures of tenebrionids were made ...". As the beetles were live-trapped and released it is quite likely that some were captured more than once, so the impressive number of 63,291 more properly refers to captures rather than individuals.

Line 286: Please provide a little more background on Ward's method. Here, and at line 343, the method appears to refer to a clustering procedure, but a citation to Ward or presentation of more detail on the method would be helpful.

Author Response

Thank you for the appraisal and helpful suggestions. The minor comments (bulleted) are followed by my responses (sub-bullets)

  • I noticed only one typographical error: at line 23 'explains' should be 'explain'
    • corrected
  • Lines 50-53: In stating that "Relatively few studies have been concerned with animal populations that bridge interpulse periods while maintaining the capacity to irrupt in response to sudden increases in water and food availability," I suggest this is correct for invertebrates but not for all desert animals. There have been, for example, quite a number of long-term studies of birds, mammals, frogs and reptiles in global desert environments, with many of these encompassing the reserve phase and showing marked population irruptions after heavy rainfall. Perhaps some of these studies could be cited, or the sentence reworded to reflect that the statement refers primarily to invertebrates.

    • The reworded sentence cites two books (Ward 2016 and Dean 2004): Notwithstanding records on vertebrates such as birds, mammals, frogs and reptiles [5, 6], relatively few studies have been concerned with arthropod populations that bridge interpulse periods while maintaining the capacity to irrupt in response to sudden increases in water and food availability. 
  • Lines 169-171: Pitfall traps were monitored twice or thrice weekly and beetles released near the site of capture. Thus, beetles could be in traps for 2-3 days before being checked. Is there any chance that the recorded beetle numbers could have been affected by predators visiting the traps and removing beetles between checks? If predation was a constant, this would not be an issue. But if predators such as mobile birds or mammals increased in response to the pulses of rainfall, could these have temporarily increased predation pressure on the trapped beetles and introduced bias in the numbers that were counted in the traps? I suspect that any such biases may wash out over 45 years, but the author may wish to discuss whether predators were present and could have had potential effects.

    • I inserted the sentence: "Solifugids, scorpions, spiders, lizards, and geckos, amounting to 4.3% of captures, could have preyed on trapped beetles, as could have transient birds and gerbils, but their impact on the results is unknown, and is assumed not to have biased the relative records of tenebrionid species over time."  
  • Line 218: Here and later in the results, please clarify what ± refers to. I presume it is standard deviation, but it is worth specifying. 

    • all cases were amended to read: "±SD"
  • Line 251: I suggest rewording "A total of 63,291 tenebrionids were captured" to something like "In total, 63,291 captures of tenebrionids were made ...". As the beetles were live-trapped and released it is quite likely that some were captured more than once, so the impressive number of 63,291 more properly refers to captures rather than individuals.

    • The reworded sentence reads: In total, 63,291 captures of tenebrionids were made at a rate of 0.31 ±SD 0.43.trap-1.day-1 (annual range 0.0016-2.4644.trap-1.day-1). 
  • Line 286: Please provide a little more background on Ward's method. Here, and at line 343, the method appears to refer to a clustering procedure, but a citation to Ward or presentation of more detail on the method would be helpful.

    • as mentioned in my response to Referee-2, the sentence in L203 has been reworded to explain this as follows: "Abundance fluctuations were tested for cyclicity by autocorrelation (lagged 3-22) of log(N+½) and GC. Finally, species were compared by correlation (Pearson’s r), which formed the basis of distance measures of a cluster analysis, using Ward’s method of hierarchical grouping attained by minimising Sum of Squares [71]. Species that correlated and clustered most closely (P<0.001) were assigned to distinct groups."